# Patients' experiences with fluctuations in persistent physical symptoms: a qualitative study

Hieke Barends [1,2] Ella Walstock,[1,2] Femke Botman,[1,2] Anja de Kruif,[3] Nikki Claassen,[1,2] Johannes C van der Wouden,[1,2] Tim olde Hartman,[4,5] Joost Dekker,[2,6] Henriette van der Horst[1,2]

For numbered affiliations see end of article.

**Correspondence to**
Hieke Barends;
h.barends@amsterdamumc.nl

## ABSTRACT

**Objectives** To explore patients' experiences with fluctuations in persistent physical symptoms (PPS) and to understand which factors—from their viewpoint—play a role in these fluctuations.

**Design** Qualitative study using semistructured interviews and thematic content analysis.

**Setting** This qualitative study is part of a multicentre prospective cohort study on the course of PPS. Patients were recruited in general practices and specialised treatment facilities for PPS throughout the Netherlands.

**Participants** Interviews were conducted with a sample of fifteen patients with PPS to explore their experiences with fluctuations in symptom severity.

**Results** We identified three themes in the analysis: (1) patterns in symptom fluctuations (2) perceived causes of symptom exacerbations and (3) Patients' strategies in gaining control over symptom exacerbations. Daily and weekly fluctuations in symptoms were an important element in patients' experiences. In particular anticipating on the worsening of symptoms impacted their daily routines and posed various challenges. Symptom exacerbations were attributed to overstepping physical limits and/or the impact of negative emotions. Resigning to physical limits, adjusting ones daily planning, weighing personal needs and learning to say 'no' were described as different strategies in gaining control over symptom exacerbations.

**Conclusions** Fluctuations in the severity of symptoms—and in particular daily and weekly symptom exacerbations—are an important element of the symptom experience in patients with PPS and poses various challenges. Patients attributed symptom exacerbation to overstepping physical limits and/or negative emotions. Patients described different strategies in gaining control over symptom exacerbations.

## INTRODUCTION

Patients with physical symptoms not attributable to verifiable, conventionally defined diseases are common in all medical settings. These symptoms are often referred to as 'medically unexplained symptoms' (MUS). A recent and perhaps more appropriate term—putting less emphasis on the mind-body dualism in the origin of symptoms—is

### Strengths and limitations of this study

► Qualitative research was applied to understand patients' experiences with fluctuations of symptoms and factors playing a role in these fluctuations from their perspective.

► Our study highlights that fluctuations in the experienced severity of symptoms—and in particular daily and weekly symptom exacerbations—are an important element of the symptom experience in persistent physical symptoms (PPS) and deserve more attention in care for these patients and in research.

► Patients were recruited in general practices as well as in specialised PPS programmes in different parts of the Netherlands, and in that regard represent a broad sample of patients with PPS.

► All of the recruited patients experienced (episodes of) severe PPS and most experienced symptoms for an extensive period of time (>5 years), therefore, our findings may be less applicable to patients experiencing mild or moderate symptoms or symptoms of short duration.

persistent physical symptoms (PPS).[1 2] When these symptoms persist, they can have a severe impact on patients' quality of life and functional capabilities and also on society due to high medical care utilisation and loss of productivity.[3 4]

There has been extensive debate about definitions and terminology in this field of research. Whereas some emphasise commonalities and overlap in symptoms and characteristics,[5–8] others differentiate between particular functional somatic syndromes (FSS), such as fibromyalgia, chronic fatigue syndrome (CFS) and irritable bowel syndrome.[9–11] The importance of studying both similarities as well as differences has also been highlighted.[12] In this study, we focus on similarities and overlap in patients' symptom experiences. We defined PPS as symptoms, which last at least several weeks and for which no sufficient somatic explanation is found

after proper medical examination by a physician. This is in line with the current Dutch multidisciplinary and general practice guidelines for MUS (PPS).[13][14] So, by definition our umbrella term PPS may also cover several FSS.

Fluctuations in symptoms have been described in several quantitative studies in patients with FSS.[15–17] Most studies on the course of PPS in a broad sample of patients used a single follow-up measurement in time to determine improvement or deterioration. According to a number of studies conducted in primary and secondary healthcare settings, 50%–75% of patients with PPS showed symptom improvement over time, whereas 10%–30% worsened.[18] In a cohort study that we conducted on the course of PPS we found improvement (63%) and deterioration (27%) rates that were in line with prior literature, when using total changes scores based on two measurements. However, when four available measurements were taken into account, the temporal stability of these outcomes was limited, as intrapatient fluctuations were highly prevalent.[19] These findings suggest that most patients with PPS might experience exacerbations and remissions in symptoms.

To the best of our knowledge, no prior qualitative study focused specifically on fluctuations of symptoms in PPS. Understanding the experiences of fluctuations in symptom severity may help medical professionals in providing care for these patients. This knowledge may enable them to understand what their patients are dealing with and to provide better guidance and support to patients with PPS. Therefore, the aims of this qualitative study were to explore patients' experiences with fluctuations in the severity of symptoms and—if present—to gain insight into factors influencing fluctuations in their symptoms from the patients' perspective.

## METHODS
### Study design
The present study was part of a larger prospective cohort study that monitors the course of symptoms and physical functioning in patients with PPS. We chose a qualitative design and conducted semistructured (in-depth) interviews, to obtain information about the experiences of patients with PPS.

### Patient and public involvement
Patients or the public were not actively involved in the design, conduct, reporting or dissemination plans of our research.

### Participants
Participants were selected from the PROSPECTS study, a PROSpective cohort study on prognosis and PErpetuing faCTors of MUS (see box 1). For the PROSPECTS study, patients filled in questionnaires about the nature and severity of their symptoms (Patient Health Questionnaire-15 (PHQ-15), 0–30 scale[20]) and physical functioning

---

**Box 1  The PROSPECTS study**

The PROSPECTS study is a Dutch longitudinal cohort study following patients (n=325) with persistent physical symptoms (PPS). PPS patients aged between 18 and 70 years were recruited in general practices (n=218) and in specialised PPS programmes of secondary and tertiary care organisations (n=107) across the Netherlands in 2013–2015. Initially patients were followed over a period of 3 years with five measurements in time (baseline, 6, 12, 24, 36 months of follow-up).[32] In 2017, the follow-up period was extended to a period of 5 years, adding a 48 and 60 months follow-up measurement. Baseline characteristics and information on the recruitment process and first 2 years of follow-up have been published elsewhere.[19][33] Over a 3-year period, only a minority of the participants (<15%) showed clinical stability in symptom severity and physical functioning.

Definition of PPS: PPS was defined as the presence of physical symptoms, which had lasted at least several weeks and for which no sufficient explanation was found after proper medical examination by a physician. This is in line with the current Dutch multidisciplinary and general practice guidelines for medically unexplained symptoms.[13][14]

---

(Research and Development-36 (RAND-36) Physical Component Summary (PCS), 0–100 scale[21]) among other questionnaires. We wanted to include patients with fluctuations as well as patients with a (seemingly) stable course of their PPS, because symptom experiences in terms of stability and fluctuations might differ between these patients. Therefore, we selected patients who: (1) showed either clinically relevant fluctuations or clinical stability (based on minimal clinically important differences) in symptom severity (PHQ-15) and physical functioning (RAND-36 PCS) over a 3-year time period and (2) had given informed consent to be contacted for future research. We used purposive sampling to ensure a diversity of participants in terms of nature of symptoms, age, gender, social characteristics (educational level, living in a rural/ urban area) and recruitment setting. Over a 3 years period, only a minority of the participants (<15%) showed clinical stability in symptom severity and physical functioning.

Patients were approached by phone by HB or EW. In total, 21 patients were contacted. Two patients were not willing to participate because of personal reasons, three patients refused because of time constraints. One patient cancelled the interview appointment due to work-related reasons. All selected patients provided written informed consent.

Fifteen patients agreed to participate. All of the recruited patients experienced (episodes of) severe PPS and most experienced symptoms for an extensive period of time (>5 years). Nature of symptoms varied. Almost all of them (n=14) had symptoms in at least two of the following symptom clusters: (1) gastrointestinal; (2) cardiopulmonary; (3) musculoskeletal and (4) general symptoms (headache, dizziness, memory impairment, concentration difficulties, fatigue). These symptom clusters were identified in a prior study by Fink *et al*[7] and are also used in the Dutch general practice guideline

**Table 1** Patient characteristics

| Variable | (n/15) |
|---|---|
| **Fluctuations/stability** | |
| Fluctuations in SS and PF | 9/15 |
| Stable in SS and PF | 5/15 |
| Fluctuations in SS, stable in PF | 1/15 |
| **Symptoms** | |
| Fatigue | 12/15 |
| Musculoskeletal pain | 12/15 |
| Headache | 6/15 |
| Gastrointestinal symptoms | 5/15 |
| Cardiopulmonary symptoms | 3/15 |
| Dizziness | 3/15 |
| Mean age (years, range) | 55.4 years (range 32–73 years) |
| **Gender** | |
| Male | 3/15 |
| Female | 12/15 |
| **Education** | |
| Higher educational level | 4/15 |
| Intermediate educational level | 4/15 |
| Lower educational level | 7/15 |
| **Living area** | |
| Rural area | 5/15 |
| City | 10/15 |
| **Recruitment setting** | |
| General practice | 12/15 |
| Specialised PPS programme | 3/15 |

PF, physical functioning; PPS, persistent physical symptoms; SS, symptom severity.

for MUS.[13] A substantial number of patients (n=10) had symptoms in at least three of these symptom clusters. Details on experienced symptoms and other characteristics of the patients are shown in table 1.

## Data collection

Interviews took place between January and April 2019. Based on the preference of the patient, 11 interviews were conducted at the patients' home and 4 at the research department of the university in a private meeting room. All interviews were digitally recorded. The interviews took 60 min on average (range: 33–93 min). Patients received a gift voucher of €15. Participants were told that the main interviewer (HB) is a general practitioner (GP) registrar and researcher with an interest in PPS and the fellow interviewer (EW) a medical intern involved in a research project on PPS. Both interviewers are female. HB had received training in qualitative research and was supervised by an experienced qualitative researcher (AdK).

Interviews were loosely structured using a topic guide with relevant areas explored in depth. The main interviewer (HB) emphasised that that all interviews were non-judgmental, confidential and anonymised. She also told the participants the researchers were particularly interested in the course of their symptoms over shorter (days, weeks) and longer (months, years) periods of time. The topic guide consisted of five main topics: (1) the experienced course of symptoms and how symptoms interfered with their daily activities, with special focus on stability and fluctuations over time (day, week, month, year(s)) ; (2) factors contributing to fluctuations in symptoms ; (3) management of symptoms and fluctuations ; (4) the role of their social and work environment and (5) the role of the healthcare system and care providers.

Based on our prior quantitative study,[19] our preconception was that patients might experience fluctuations in symptoms and that these might be relevant to them. Based on theoretical sampling, we selected 'fluctuating' as well as 'seemingly stable' patients. We expected more prominent accounts on fluctuations in the 'fluctuating' patients. While we had this preconception, we asked open questions in both 'fluctuating' as well as 'seemingly stable' patients about the experienced symptoms over time (a day, a week, a month, etc) when interviewing the patients.

Patients were encouraged to talk freely about their experiences and expand on any aspects they felt were relevant. The topic guide was checked throughout the interview process, no major adjustments were made. All participants received a summary of the interview afterwards for a member check. Fourteen patients responded to the summaries, they confirmed that they recognised their experiences in the summaries and no major changes in content were made.

## Data analysis

All interviews were transcribed verbatim and coded using Atlas.ti vV.7. The analysing process was based on thematic analysis according to the six phases described by Braun and Clarke.[22] In all phases, at least two researchers were involved (HB, EW and FB) to enrich the analysis. In the first phase, HB, EW and FB familiarised themselves with the data by summarising and close reading. In the second phase, HB, EW and FB all read and coded the first two interview transcripts, using open coding. Codes were discussed to reach agreement and to improve internal validity. This resulted in an initial code list that was extended when further transcripts were analysed in pairs following the same strategy. In the following phases, codes were clustered into subthemes in order to identify patterns in the interviews, after which final themes were identified. HB, EW and FB discussed codes, subthemes and themes until consensus was reached on all themes. Constant comparison was used in order to understand differences and similarities between patients and within each patient. All results were discussed in the research team to enhance the robustness of the findings. Finally, the report was produced and quotes were extracted that related to the themes. We used the Standards for

Reporting Qualitative Research checklist when writing our report.[23]

## RESULTS

Three main themes were identified in the analysis: (1) Patterns in symptom fluctuations, (2) Perceived causes of symptom exacerbations and (3) Patients' strategies in gaining control over symptom exacerbations.

### 1. Patterns in symptom fluctuations

All interviewed patients experienced fluctuations in the occurrence and severity of symptoms. This meant that both the selected patients with fluctuations, as well as the seemingly stable patients in our sample experienced fluctuations.

#### Short-term fluctuations

Fluctuations in symptoms occurred in particular over the day, but also over the week.

> And it varies. One day I am in the shower and I think 'Here it comes'. The next day, well, it can start during the day. And sometimes, very occasionally, I will be fine. (P4, female)

Most patients experienced a gradual worsening of symptoms over the day and work week. Others did not experience a specific pattern. Worsening over the work week was described by all patients who worked.

> If I wake up with little pain, it is a good day. But a day will eventually always end with pain. (P3, female)

> At the end of the week it is usually worse. (P8, male)

Only few patients did not experience a recognisable pattern over the day or week.

#### Long-term fluctuations

Most patients described exacerbations and remissions of symptoms and how these symptoms influenced their lives over longer periods of time (months-years).

> And I've also had periods when I was able to do other things as well. So there have been periods when things were better, and I could do a little more. (P2, male)

Throughout their lives, a couple of patients described several isolated episodes of symptom exacerbations that lasted at least several months, as well as periods that had been free of symptoms. At the time of the interview, some patients reported a recent increase in symptoms over the weeks before the interview, whereas one patient was free of symptoms at the time of the interview. In some improvement was present, but only for relatively short periods.

> Well, yes, there are bad days and good days, but then there are more bad ones. (P12, female)

In particular for short-term fluctuations patients indicated to continuously search to understand and explain what caused the exacerbations of symptoms, so they could anticipate on and prevent symptoms from worsening.

### 2. Perceived causes of symptom exacerbations

#### Overstepping physical limits

Patients described an increase in symptom severity when overstepping their physical limits. Overdoing it was experienced as leading to setbacks with exacerbations of symptoms. Many patients therefore aimed for a certain balance: a balance between their aims and abilities, pushing physical limits but not overdoing it.

> At first I was up and down, all over the place. I really thought 'I'll get over this, I'll do it again, I'll do everything again (…) Well, it takes a couple of years before you really hit the wall and think 'sorry, you can try as hard as you like, you will still have these setbacks.' And then you can start all over again, because then you are overstepping your limits. (P11, female)

Some patients described an energy balance. In case of a negative balance, symptoms worsened. Many patients experienced a link between this energy balance and the progression of their symptoms during the day or week.

> You know, it's like 'everybody has an energy span, a range of ability, that is different for every person, and you always want something else'. Only I am usually just confronted with the consequences of this sooner. Because when I think 'I'll keep going now', I'll have a problem tomorrow. (P3, female)

Almost all patients mentioned the importance of respecting their physical limits in order to prevent their symptoms from worsening and to experience fewer fluctuations. Some patients also mentioned the importance of staying active and searching for the right balance, as not doing enough also resulted in worse symptoms in these patients.

> It is 'I did either too much, or not enough'. One or the other. (P11, female)

#### Negative emotions

A couple of patients experienced that their symptoms represented or were exacerbated by negative emotions. One patient linked her symptoms solely to negative emotions and viewed her symptoms as a representation of these emotions. She found the solution in getting a hold of her emotions—that she attributed to her personal situation at that time. By changing her personal situation with the help of her religion, she explained she got rid of these negative feelings. At the time of the interview, she was free of symptoms.

> Well, you know, you are angry, you are sad, or a little depressed. (…) Why is this happening to me? (…) But the physical pains I sometimes had, that was purely because I was sad. You know, that stress that sometimes enters your system. And it also has to do

with resignation. How much of your situation do you accept? (P5, female)

Others mentioned how their symptoms led to worries and negative feelings, and that from their perspective these feelings worsened the symptoms and created a vicious circle.

At that moment I thought I was dying. And then you get stressed. That's what happens. Then you are in more and more pain. So eventually you get into this vicious circle as a human being. Because when you start thinking 'yes, it is indeed getting worse', that's what happens. (P1, female)

The effect of emotionally stressful events not related to their symptoms was also mentioned as resulting in symptom exacerbations. In these cases patients felt not capable of taking control over their symptoms.

In an event like that, I won't be able to sit down. There is too much adrenaline in my system. A whole lot of symptoms will follow. Not directly, but after a day or two, when things are calming down a bit. (P3, female)

We were having a good time together, but then my granddaughter suddenly started to bark at us, she is hitting puberty you know. That hurts. (…) Than you can feel it in your shoulders, you know, because your muscles get more tense. (P12, female)

For some patients it was difficult to acknowledge a relation between negative emotions and their physical symptoms, although they believed there was some connection. One patient with a recent increase in symptoms mentioned the following on this:

R: Yes, if I'm being honest to myself, I think that it [negative emotions due to job loss] got in my way.

I: Are you experiencing more symptoms since then? Do you connect this?

R: Well, I don't exclude it. (…) When you're honest, I know that myself, you know that it probably plays a role. (P8, male)

This view followed after a somatic disease was excluded by the GP and several medical specialist over the last couple of months.

You can no longer exclude it, when you are physically healthy. (P8, male)

Although a couple of other perceived causes of symptom exacerbations were mentioned (sleep disturbances; focusing on symptoms; food allergies)—these did not have a prominent role in patients' personal accounts and were mentioned as having some impact in addition to the prominent impact of physical limits and/or negative emotions.

## 3. Patients' strategies in gaining control over symptom exacerbations

### Resigning to physical limits

Patients mentioned the importance of respecting their physical limits in order to prevent their symptoms from worsening and to experience fewer ups and downs. They experienced that ignoring their symptoms and not taking their physical limits into account resulted in symptom exacerbations. The lack of recognition and validation of symptoms in the absence of a diagnosis or plausible explanation was mentioned as creating difficulties in respecting and resigning to personal physical limits.

Only, realistically, I sometimes think 'well, but I don't have a problem'. 'There is nothing wrong with me'. Everything is in working order, so I should be able to just do that. Often this is what gets in the way, like 'it's all in my mind'. You know, why not push through? But then I immediately pay the price. (P3, female)

Although many patients mentioned the importance of resigning to physical limits, they described this ongoing process as challenging and often frustrating. By resigning, we mean that patients expressed the need to take their physical limits seriously and anticipate by limiting their activities in order to prevent exacerbations of symptoms. Resigning to limits was experienced as different from accepting their limits, as many kept struggling with the acceptance of their physical limits. They for example encountered new situations as a result of changing environments and life changes over time, again confronting them with their physical limits.

I still haven't fully embraced it and am not Zen about it. Because, you know, when I see other mothers. (…) Or when Mum plays tag or something. Then I run ten paces. Can't run too long, or I get myself in trouble. That still frustrates me. (P3, female)

Finding a plausible explanation for the experienced symptoms was seen as helpful in accepting their physical limits. In case an explanation was offered and made sense to the patient, it contributed to the understanding of their symptoms and helped them in acceptance of limitations.

Well, that was really good, because then you finally have an explanation for the symptoms. Because she could also explain where this fatigue comes from. And then, well, you adjust your life to it. So I accepted it, that I can just do less. (P2, male)

### Adjusting daily planning

Most patients eventually adjusted their daily planning and routines to their physical limits and capabilities. They mentioned pacing activities and resting effectively as important strategies in gaining control over symptoms and experiencing fewer ups and downs.

The entire day I keep in mind what I need and want to do. So if I have a birthday tonight, than I take a nap in the afternoon. (P13, male)

In some patients, incorporating mindfulness and relaxation exercises into their daily routine had positive results regarding their experienced symptoms.

### Weighing personal needs and learning to say 'no'

Other patients indicated to have gained control by continuously weighing personal needs. Deciding to participate in joyful activities, while they knew it would exacerbate their symptoms, helped some patients to cope. They described to weigh the personal gain and consequences and in some cases decided to consciously overstep their limits, anticipating an exacerbation of symptoms.

Yes, and also that you know when you are overdoing it and you still choose to do that, knowing that you will be in serious pain the next day. That makes it easier to accept. The harder you fight, the angrier, I think, you will get and the worse your pain will be. (P11, female)

Whereas in some cases the activity was worth overstepping limits, in other cases patients thoughtfully evaluated the activity as having too little value to them. Learning to say no, in such cases, was also experienced as important in gaining control over symptoms.

And sometimes I think: I am not going to do that. If I am not well and it's not something I really enjoy. No … I evaluate: is it worthwhile, does it do me any good? Is it something I enjoy? If not, I say no. You also need to learn to say 'No'. I didn't do that when this started. (P12, female)

### DISCUSSION

Our findings highlight that fluctuations in symptoms—and in particular the symptom exacerbations that patients describe over the day and week—are an important element of symptom experience in patients with PPS. It impacts their daily routines and poses various challenges. Patients attributed the experienced worsening of symptoms in particular to overstepping their physical limits and/or to the impact of negative emotions. Patients described different strategies in gaining control over symptom exacerbations: by resigning to their physical limits, adjusting their daily planning to their limits and capabilities, weighing personal needs and learning to say 'no'.

As far as we are aware, this is the first qualitative study exploring the experiences with fluctuations in symptoms among patients with PPS. A strength of this study was the fact that patients were recruited in different healthcare settings throughout the Netherlands and that patients varied with regard to diversity of symptoms and demographic and social characteristics. More female patients were interviewed, but numbers were in line with the balance in the cohort from which we selected the patients (75% female). We tried to minimise the impact of our preconceptions, for example, by our theoretical sampling method in which we included both 'fluctuating' and 'seemingly stable' patients. Although we anticipated differences in experience between these patients, this was in fact not the case. A limitation of our study is that all interviewed patients experienced (episodes of) severe PPS for longer periods of time, hence our findings may be less applicable to patients experiencing mild symptoms or symptoms of short duration.

Our findings correspond to some findings from quantitative studies in CFS. Ecological momentary assessment established that patients experienced difficulties in balancing their activities in response to symptoms. More fatigue-related symptoms and pain predicted more activity limitation whereas feeling subjectively well predicted more all-or-nothing behaviour, resulting in ups and downs.[15] Pacing activities was helpful in preventing fluctuations in symptoms.[16] Comparable quantitative studies in a broader sample of patients with PPS are currently lacking. Our findings, however, suggest that dealing with fluctuations—and in particular anticipating on symptom exacerbations—seems to apply to the broader spectrum of PPS.

In our study, resignation to physical limits was mentioned as a strategy to anticipate on and prevent symptom exacerbations. Having a plausible explanation for symptoms was helpful in acceptance of experienced physical limitations. A prior qualitative study described that in particular patients who displayed acceptance of PPS—as opposed to resignation—shifted their focus towards improving their quality of life.[24] Resignation and acceptance seem closely related, but the latter implies to be a later stage in a process of change. Acceptance was also an important condition for symptom improvement[25] and facilitated a process of change towards self-compassion and self-care in patients with PPS.[26]

While in our study, resigning to limits was described as an important strategy in anticipating on symptom exacerbations and fluctuations in symptoms, Sowińska and Czachowski[27] described how in their population of Polish patients with PPS (MUS) ignoring symptoms or shifting away attention was reported as one of the most successful ways of coping. These differences are interesting. The Polish patients are likely to represent a different selection of patients with PPS: they were all included in the same general practice and visited psychologists and psychiatrist privately. Cultural differences may play a role as well. Multiple studies[24 28 29] highlight PPS patients' concerns that it might be 'all in the mind' and how this often brings shame and the feeling of not having 'a legitimate illness'. In our study, patients also struggled with their physical limits in the absence of a 'legitimate illness'. Although symptom exacerbations were attributed to negative emotions by some patients in our study, patients also indicated initial difficulties in accepting the connection. In a

recent study on consultations between GPs and patients, symptoms could be attributed to emotions when patients introduced this link themselves. However, when the GP introduced this link it tended to be denied.[30] This again underlines the stigma that still pertains on mental distress and its relation to physical health.

Several of our findings may be helpful in the care for patients with PPS. First, our study again underlines the need to take symptoms and their consequences seriously as a healthcare provider (HCP), also in the absence of an identifiable disease. Patients with PPS face challenges in dealing with fluctuations in symptoms, and more specific in dealing with symptom exacerbations. Second, as an HCP, exploring patients' experiences with symptom exacerbations—with attention paid to the experienced impact of physical limits and negative emotions—might be a useful starting point to gain an understanding of what your patient is struggling with on a daily basis and may create a common ground for supportive care to improve well-being and provide illness-based interventions and advice.

Our study highlights that fluctuations in symptoms are an important element of the experienced symptoms. More longitudinal research into short-term fluctuations in experienced symptoms in a broad sample of patients with PPS, for example, by the experience sampling method (ESM), could provide useful new insights. ESM can reveal how symptom experience relates to implicit patterns of thought, experience and behaviour.[31] Another valuable area of research could be the different strategies of gaining control over symptom exacerbations and their impact on functional health and well-being.

**Author affiliations**
[1]Department of General Practice and Elderly Care Medicine, Amsterdam UMC - VUMC location, Amsterdam, Noord-Holland, The Netherlands
[2]Amsterdam Public Health Research Institute, Amsterdam, Noord-Holland, The Netherlands
[3]Department of Health Sciences, Faculty of Science, Vrije Universiteit Amsterdam, Amsterdam, The Netherlands
[4]Department of Primary and Community Care, Radboud University Nijmegen Medical Center, Nijmegen, Gelderland, The Netherlands
[5]Donders Institute for Brain Cognition and Behaviour, Radboud University, Nijmegen, Gelderland, The Netherlands
[6]Department of Rehabilitation Medicine and Department of Psychiatry, Amsterdam UMC - VUMC location, Amsterdam, Noord-Holland, The Netherlands

**Acknowledgements** We want to thank all the patients that took part in this study for sharing their experiences.

**Contributors** HB, NC, JCvdW, JD and HvdH developed the study protocol. AdK and ToH provided feedback on the study protocol. HB and EW contributed to the development of the topic guide, collected and analysed the data and interpreted the results. FB analysed the data and interpreted the results. AdK provided feedback on the data collection and analysis. HB drafted the manuscript. All authors read, provided critical revisions and approved the manuscript.

**Funding** This work was supported by grants of ZonMw [funding number 839 110 018] and the Stoffels-Hornstra Foundation. The funding sources had no role in the design, analyses and interpretation of this study.

**Competing interests** None declared.

**Patient consent for publication** Obtained.

**Ethics approval** The institutional review board of the Amsterdam UMC (IRB00002991) approved the research protocol (No. 2018.483).

**Provenance and peer review** Not commissioned; externally peer reviewed.

**Data availability statement** No data are available.

**ORCID iD**
Hieke Barends http://orcid.org/0000-0002-5480-0017

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
