## [Reviewer comments · BMJ Open]

ARTICLE DETAILS

TITLE (PROVISIONAL)	Patients' experiences with fluctuations in persistent physical symptoms: a qualitative study
AUTHORS	Barends, Hieke; Walstock, Ella; Botman, Femke; de Kruif, Anja; Claassen, Nikki; van der Wouden, Johannes; Olde Hartman, Tim; Dekker, Joost; van der Horst, Henriette

VERSION 1 - REVIEW

REVIEWER	Caroline Mitchell University of Sheffield, UK
REVIEW RETURNED	11-Dec-2019

GENERAL COMMENTS	Overall this is a good qualitative study with attention paid to fulfilling rigor in standards of conduct of research. Participants were positively selected and interviews continued to data saturation . Data analysis : term 'reliability' is not usually used in reference to independent verification of themes in qualitative research - cf Lincoln and Guba criteria - please rephrase (line 34 page 7 "
---

REVIEWER	Ditte Roth Hulgaard University of Southern Denmark Denmark
REVIEW RETURNED	12-Feb-2020

GENERAL COMMENTS	Dear authors, First off, I want to congratulate you on your wonderful PROSPECTS study, which is relevant and needed in the field of PPS. Further I applaud the choice to use both quantitative and qualitative methods in order to broaden perspectives and knowledge. The current study has a relevant focus and appropriate methodology. However, the paper needs substantial work with regard to qualitative rigor, analysis and interpretation. Thus, I have minor comments for the introduction and methods section, while results and discussion need a substantial revision. Introduction:
--

P4 line10: Please elaborate a little how symptoms may vary in severity and how severity may impact impairment.
P4 line13: Definitions and terminology are much debated in this area. Please define more clearly – are all FSS also PPS – and vice versa?
P4 line 19 – 21: The studies you are referring to are on MUPS, somatization and hypochondriasis. Are all of these covered under your definition of PPS?
The definition of PPS from Box 1 might be helpful in the introduction, with an elaboration on how it relates to other terms used in the paper.

Method:

Overall the context, sampling strategy and data collection are well described.

Box1: very helpful elaboration of the overall study.

I have only a minor comment for this section:

P5 line 17: Please elaborate the definition of symptom clusters

P 7: The sections on data-analysis, the description of reflexivity, analysis and trustworthiness need more work.

First off, the research question in itself suggests that researchers entered this project with a hypothesis, or preconception if you want, that fluctuations in symptom severity are of importance for patients with PPS. This however is also one off the main conclusions. And while this can be the case, some consideration of what preconceptions were and how these may have influenced the analysis is relevant.

In the methods section, consensus decision and reliability are mentioned. However, reliability as defined in quantitative research cannot be applied directly in a qualitative study. Please elaborate how reliability is defined and how the validity of findings is supported. You might find inspiration in the following papers: Malterud, K. (2001). Qualitative research: standards, challenges, and guidelines. *Lancet*, 358(9280), 483-488. doi:10.1016/s0140-6736(01)05627-6; Malterud, K. (2001).

The art and science of clinical knowledge: evidence beyond measures and numbers. *Lancet*, 358(9279), 397-400. doi:10.1016/s0140-6736(01)05548-9

Results:

The results section needs substantial work. Again, you might want to draw on the two papers by Malterud for inspiration.

Throughout the results section there is emphasis on commonalities, consensus and on “what most patients say”. However, in a qualitative inquiry, focus should also be on different perspectives on a theme, on understanding the variations in how fluctuations are experienced and influenced rather than the mere statement that they are. This could be kept in mind throughout the revision process.

Further the results section lacks qualitative rigor. The authors present their own thoughts and interpretations, sometimes conclusions, between quotations, but there is not always sufficient support for the statements made in the quotations used.

Quotations should support the statements made and the reader should be able to follow the researcher’s line of thought.

I will provide a specific example:

P 9 line 47, crossing emotional boundaries:

A number of patients mentioned not just the importance of

respecting their physical limitations, but also the importance of respecting their emotional boundaries.

“And sometimes I think: I am not going to do that. If I am not well and it’s not something I really enjoy. No ... I evaluate: is it worthwhile, does it do me any good? Is it something I enjoy? If not, I say no. You also need to learn to say ‘No’. I didn’t do that when this started.” (P12, female, 70 yrs)

A couple of patients indicated that their symptoms also worsened following acute emotionally stressful events. In these cases, they felt not capable to control symptoms.

For me as the reader, the chosen quotation neither supports the above nor the below statement. If it does so, the authors don’t succeed in explaining how, sufficiently, for me to understand. The patient describes learning to say “no”. Saying no however, can be a way to respect personal physical boundaries rather than emotional boundaries. Further the quote also suggests that saying no depends on other factors than possible symptom exacerbation. The patient describes how she evaluates potential activities based on possible personal gain and enjoyability. So it is not so much about avoiding “overstepping of boundaries”, rather it is about gaining something, setting priorities. Further, it is very interesting to learn how P12 has acquired the skills to say “no”, -is she more aware of what is good for her, is she less afraid of possible consequences...? However, the quote doesn’t say anything about this – but maybe that could be an interesting theme on its own? I don’t have sufficient context to provide a meaningful analysis of a single quotation off course, so this should just be seen as inspiration.

The above general comments are relevant throughout the results section.

Below, I will present some more specific comments to individual subthemes.

P8: Theme 1 – the first section lists a number of quotations which support what you already knew from your sampling - that you have included patients who do in fact experience fluctuation. So this is hardly an interesting or surprising finding.

The quotations, however, also pertain to “patterns of fluctuation” – this might be a slightly different focus to explore in this theme?

P8 line 12: in your presentation of participant selection you emphasize the importance of selecting participants with fluctuating and stable symptoms respectively. Please elaborate how that relates to the fact that all participants in fact do experience fluctuation and how this may affect your findings.

P8 line 21: paid or unpaid jobs, what do you mean? Does it matter?

P 8 line 45: In these final quotes, patients talk about how the anticipation of symptom exacerbation, rather than experienced fluctuations, affect their daily life and activities. This may be a very relevant theme to explore. However, it does not fit in the title “experiences with fluctuation”.

P9: Theme 2. This theme addresses the patient experience of why symptoms may fluctuate, which is very relevant. Maybe the title could reflect this. The introduction provides a very interesting and relevant conclusion and may be placed at the end of the subtheme

description. The results have to support the claims though. The section on “balancing physical limits” contains relevant quotes which support the statements. The section on “crossing emotional boundaries” needs more work. As highlighted above, I cannot follow how quote 1 and 2 pertain to emotional boundaries.

P10 line 12: Quote 3 in this section is concerned with the association of negative emotions and symptom experience. I don’t know if it is about emotional boundaries? However, the influence of negative emotions on symptom experience is extremely interesting and could be elaborated further.

P9: Theme 3: Dealing with fluctuations: experienced difficulties in balancing limits and boundaries
The theme is relevant and interesting; however, the introducing statement or conclusion again does not have sufficient support in the quotations and interpretations. Overall, I think that the quotes rather are concerned with the patients’ ways of dealing with symptoms, priorities and life as such, rather than with dealing with fluctuations.

P10: Lack of recognition and validation of symptoms
The experiences with lack of diagnosis are well described in the qualitative literature and an important issue to address.

Inspiration (maybe relevant in the discussion?):

- Salmon, P. (2007). Conflict, collusion or collaboration in consultations about medically unexplained symptoms: the need for a curriculum of medical explanation. *Patient Educ Couns*, 67(3), 246-254. doi:10.1016/j.pec.2007.03.008
- Dumit, J. (2006). Illnesses you have to fight to get: facts as forces in uncertain, emergent illnesses. *Soc Sci Med*, 62(3), 577-590. doi:10.1016/j.socscimed.2005.06.018
- Jutel, A. (2016). Truth and lies: Disclosure and the power of diagnosis. *Soc Sci Med*, 165, 92-98. doi:10.1016/j.socscimed.2016.07.037
- Jutel, A., & Nettleton, S. (2011). Towards a sociology of diagnosis: Reflections and opportunities. *Social Science and Medicine*, 73(6), 793-800. doi:10.1016/j.socscimed.2011.07.014
- Nettleton, S. (2006). ‘I just want permission to be ill’: Towards a sociology of medically unexplained symptoms. *Social Science & Medicine*, 62(5), 1167-1178. doi:http://dx.doi.org/10.1016/j.socscimed.2005.07.030

However, the claim made that lack of diagnosis/validation leads to overstepping of boundaries which again leads to symptom fluctuation is simply not one that can be supported by any qualitative data. Qualitative inquiry does not provide information about causation. Please revisit this section.

In the final section on page 11: Importance of resigning to limits and boundaries, the authors make statements about how patients have gained control over symptoms. Again, the quotes do not support the claims made. The final quote rather suggests that “not resigning into symptoms” is experienced as gaining control. The text does suggest so, however the overall conclusion you present later is that: overall – resigning is a good thing.

An interesting analysis might be the different perspectives on “gaining control” that you can find in your data. 1) “resigning”, 2) “consciously overstepping”, 3) “evaluating an activity and making choices based on personal preferences” (as was suggested in a

previous quote). 4)

I am sure you have a lot of data to provide perspectives on this matter. For the clinician, this could be very important information – gaining control is achieved differently by various patients, possible methods of gaining control were....

Discussion

I will not comment on the discussion in too much detail, as it probably will change when the results section has been revised.

P11 line 16: The concluding remarks of the results are very interesting, however not sufficiently supported as described above.
P11 line 47: I absolutely agree with the strength of and relevance of this study

P12 line 13: This section starts with the claim that letting go of the search for a diagnosis results in less fluctuation which again results in improved wellbeing. Again, careful with claiming causation.

Maybe you can find inspiration for the discussion of this point in the literature suggestions above.

Checklists comments:

2. The abstract will need changes after the revision

8. The references used are relevant and up to date. However, the list reflects focus on quantitative literature. More relevant qualitative literature might prove helpful.

A little more inspiration

- Moulin, V., Akre, C., Rodondi, P. Y., Ambresin, A. E., & Suris, J. C. (2015). A qualitative study of adolescents with medically unexplained symptoms and their parents. Part 1: Experiences and impact on daily life. *J Adolesc*, 45, 307-316.
doi:10.1016/j.adolescence.2015.10.010

- Nunes, J., Ventura, T., Encarnacao, R., Pinto, P. R., & Santos, I. (2013). What do patients with medically unexplained physical symptoms (MUPS) think? A qualitative study. *Ment Health Fam Med*, 10(2), 67-79.

- Karterud, H. N., Risor, M. B., & Haavet, O. R. (2015). The impact of conveying the diagnosis when using a biopsychosocial approach: A qualitative study among adolescents and young adults with NES (non-epileptic seizures). *Seizure*, 24, 107-113.
doi:10.1016/j.seizure.2014.09.006

9.-11. Please look to the comments on the results section above

I hope that you find the comments helpful and that they will allow you to revisit your very interesting study and data with new inspiration.

I want to thank you for allowing me to comment on your work.

Best wishes

REVIEWER	MWF van den Hoogen Erasmus MC Rotterdam
REVIEW RETURNED	18-Feb-2020

GENERAL COMMENTS	The authors have performed an original and important study on a very common medical problem. Before their study it was indeed unknown how patients perceive the fluctuations in their symptoms and I find that my patients often try to find (sometimes in vain) any correlation between the fluctuations and any behavior or other attribute. Although the authors acknowledge that their data stems from a selected group of patients with PPS and might therefore not apply to all patients with PPS, I have learned a few points and so should other healthcare providers, to improve the care for these often 'difficult to treat' patients. That said, I have a few comments. First, in the methods selection both patients with fluctuations and those with (seemingly) stable course were included, however this distinction does not trickle down in the rest of the article. Can the authors explain why, was there no two different group of patients (if so, explain table 1 and this seems to contradict the first paragraph of the results, line 14-15 page 8) or did they not differ in their discussed themes? This could help personalizing the care for PPS patients. More-over I would like to see more details about the patients. What kind of PPS do they have? The authors state it is over more domains, but to give a clear picture to readers, it would help to see that xx% have pain, xx% have fatigue, etc etc. Furthermore, can the authors give more quantification on the level of fluctuation over both long and short term. Are they really fluctuations or only perceived as such, if real, how much and is that amplitude of fluctuation related to outcome e.g. are high fluctuators more impacted on daily life than low fluctuators. Another point might be about the external validity. Can the authors elaborate more how applicable these results might be in a culturally different population than Dutch people, since I think culture and perspective of PPS are two very important issues. Can we use these data on patients from other cultures living in the Netherlands / Western Europe? Can the authors also give 1-2 extra examples on the topic of "importance of resigning to limits and boundaries"? Finally, I am not sure whether all readers have the same understanding of the term 'resignation' which is used frequently. I suppose the authors use a different wording or explain the term. In all I think the authors will contribute substantially to the medical field of PPS / MUPS if these data are published.
---

VERSION 1 – AUTHOR RESPONSE

Reviewer #1:

Reviewer Name
Caroline Mitchell
Institution and Country
University of Sheffield, UK

Please state any competing interests or state 'None declared': none declared

Please leave your comments for the authors below: Overall this is a good qualitative study with attention paid to fulfilling rigor in standards of conduct of research. Participants were positively selected and interviews continued to data saturation .

Data analysis :

term 'reliability' is not usually used in reference to independent verification of themes in qualitative research - cf Lincoln and Guba criteria - please rephrase (line 34 page 7)

Response: Thank you for your helpful suggestion, we are aware that there is considerable debate about the concept of reliability in qualitative research, we therefore replaced it by more suitable terminology. We therefore draw upon Malterud's (2001) paper.

Malterud, K. (2001). Qualitative research: standards, challenges, and guidelines. *Lancet*, 358(9280), 483-488. doi:10.1016/s0140-6736(01)05627-6; Malterud, K. (2001).

Changes (p.7 Methods, data-analysis):

-“In all phases at least two authors were involved (HB, EW, FB) to increase reliability.”

was changed into:

“In all phases at least two researchers were involved (HB, EW, FB) to enrich the analysis.”

-“...improve reliability” was changed into: “...improve internal validity”

Reviewer #2

Reviewer Name

Ditte Roth Hulgaard

Institution and Country

University of Southern Denmark

Denmark

Please state any competing interests or state 'None declared': None declared

Dear authors,

First off, I want to congratulate you on your wonderful PROSPECTS study, which is relevant and needed in the field of PPS. Further I applaud the choice to use both quantitative and qualitative methods in order to broaden perspectives and knowledge.

The current study has a relevant focus and appropriate methodology.

However, the paper needs substantial work with regard to qualitative rigor, analysis and interpretation.

Thus, I have minor comments for the introduction and methods section, while results and discussion need a substantial revision.

Introduction:

-P4 line10: Please elaborate a little how symptoms may vary in severity and how severity may impact impairment.

Response: We believe that elaborating on how symptoms may vary in severity and impact functioning goes beyond the scope of this part of the introduction –where we aim at introducing the societal relevance of PPS research in general. We do, however, pay attention to this topic in the paper.

-P4 line13: Definitions and terminology are much debated in this area. Please define more clearly – are all FSS also PPS – and vice versa?

Response: In our study we focus on overlapping experiences among patients with PPS. In order to be clear on our definition, we introduce it here -as you also suggested in your next comment- and provided some more background about the debate.

Changes (Introduction p. 4):

The following was added:

“There has been extensive debate about definitions and terminology in this field of research. Whereas some emphasize commonalities and overlap in symptoms and characteristics (Nimnuan 2001, Fink 2007, Fink 2010, Budtz-Lilly 2015), others differentiate between particular functional somatic syndromes (FSS), such as fibromyalgia, chronic fatigue syndrome and irritable bowel syndrome (Wessely & White 2004, Jones 2019, Abbi 2013). The importance of studying both similarities as well as differences has also been highlighted (White 2010). In this study we focus on similarities and overlap in patients’ symptom experiences. We defined PPS as symptoms, which last at least several weeks and for which no sufficient somatic explanation is found after proper medical examination by a physician. This is in line with the current Dutch multidisciplinary and general practice guidelines for MUS (PPS) (olde Hartman 2013, van der Feltz-Cornelis 2011). So, by definition our umbrella term PPS may also cover several FSS.”

-P4 line 19 – 21: The studies you are referring to are on MUPS, somatization and hypochondriasis. Are all of these covered under your definition of PPS?

The definition of PPS from Box 1 might be helpful in the introduction, with an elaboration on how it relates to other terms used in the paper.

Response: This was partly addressed in our response to your previous comment. Regarding the studies we refer to: in the article by Olde Hartman et al. (2009) (reference number 8, line 19), separate analyses were done for PPS (MUS), somatization and hypochondriasis of which we only used results about PPS. The other article (reference number 9, line 21) is about our own study (into PPS). We added a few words to clarify this.

Changes (introduction page 4):

We added the words in italics:

According to a number of studies conducted in primary and secondary health care settings, 50-75% of patients with PPS showed symptom improvement over time, whereas 10-30% worsened (olde Hartman 2009). In a cohort study that we conducted on the course of PPS we found improvement (63%) and deterioration (27%) rates were in line with prior literature, when using total changes scores based on two measurements.

Method:

Overall the context, sampling strategy and data collection are well described.

Box1: very helpful elaboration of the overall study.

I have only a minor comment for this section:

-P5 line 17: Please elaborate the definition of symptom clusters

Response: We added our definition of symptom clusters, which is based on a study by Fink et al. (2007) (see reference below). We discern the following four symptom clusters: 1) gastro-intestinal; 2) cardiopulmonary; 3) musculoskeletal/pain and 4) general symptoms (headache, dizziness, memory impairment, concentration difficulties, fatigue). These symptom clusters are also used in the Dutch general practice guideline for MUS (olde Hartman 2013). We additionally included the symptoms and frequencies in table 1 as this was an additional request of Reviewer#3 (see below).

Fink P, Toft T, Hansen MS, Ornbol E, Olesen F. Symptoms and syndromes of bodily distress: an exploratory study of 978 internal medical, neurological, and primary care patients. *Psychosom Med.* 2007;69(1):30-9.

Changes (Methods, page 5), we adjusted/ added the following:

“Almost all of them (N=14) had symptoms in at least two of the following symptom clusters: 1) gastro-intestinal; 2) cardiopulmonary; 3) musculoskeletal/pain and 4) general symptoms (headache, dizziness, memory impairment, concentration difficulties, fatigue). These symptom clusters were identified in a prior study by Fink et al (2007) and are also used in the Dutch general practice guideline for MUS (olde Hartman 2013). A substantial number of patients (N=10) had symptoms in at least three of these symptom clusters. Details on experienced symptoms and other characteristics of the patients are shown in table 1.”

For additional changes concerning the symptoms see table 1 on page 6.

-P 7: The sections on data-analysis, the description of reflexivity, analysis and trustworthiness need more work.

First off, the research question in itself suggests that researchers entered this project with a hypothesis, or preconception if you want, that fluctuations in symptom severity are of importance for patients with PPS. This however is also one off the main conclusions. And while this can be the case, some consideration of what preconceptions were and how these may have influenced the analysis is relevant.

Response: This is indeed a relevant point. We therefore added a few sentences on our preconceptions to the methods section.

Changes (data collection, p. 7):

The following was added:

“Based on our prior quantitative study (Claassen-van Dessel 2018), our preconception was that patients might experience fluctuations in symptoms and that these might be relevant to them. Based on theoretical sampling, we selected ‘fluctuating’ as well as ‘seemingly stable’ patients. We expected more prominent accounts on fluctuations in the ‘fluctuating’ patients. Whilst we had this preconception, we asked open questions in both ‘fluctuating’ as well as ‘seemingly stable’ patients about the experienced symptoms over time (a day, a week, a month etc.) when interviewing the patients.”

-In the methods section, consensus decision and reliability are mentioned. However, reliability as defined in quantitative research cannot be applied directly in a qualitative study. Please elaborate how reliability is defined and how the validity of findings is supported. You might

find inspiration in the following papers:

Malterud, K. (2001). Qualitative research: standards, challenges, and guidelines. *Lancet*, 358(9280), 483-488. doi:10.1016/s0140-6736(01)05627-6; Malterud, K. (2001).

The art and science of clinical knowledge: evidence beyond measures and numbers. *Lancet*, 358(9279), 397-400. doi:10.1016/s0140-6736(01)05548-9

Response: Thank you for your helpful suggestions. Reviewer #1 also commented on our use of the term 'reliability'. As we addressed in our response to reviewer#1, we are aware that there is considerable debate about the concept of reliability in qualitative research and we therefore replaced this term by more suitable terminology in qualitative research. For changes see our response to the comment of reviewer#1.

Results:

The results section needs substantial work. Again, you might want to draw on the two papers by Malterud for inspiration.

Throughout the results section there is emphasis on commonalities, consensus and on "what most patients say". However, in a qualitative inquiry, focus should also be on different perspectives on a theme, on understanding the variations in how fluctuations are experienced and influenced rather than the mere statement that they are. This could be kept in mind throughout the revision process.

Further the results section lacks qualitative rigor. The authors present their own thoughts and interpretations, sometimes conclusions, between quotations, but there is not always sufficient support for the statements made in the quotations used. Quotations should support the statements made and the reader should be able to follow the researcher's line of thought.

I will provide a specific example:

P 9 line 47, crossing emotional boundaries:

A number of patients mentioned not just the importance of respecting their physical limitations, but also the importance of respecting their emotional boundaries.

"And sometimes I think: I am not going to do that. If I am not well and it's not something I really enjoy. No ... I evaluate: is it worthwhile, does it do me any good? Is it something I enjoy? If not, I say no. You also need to learn to say 'No'. I didn't do that when this started." (P12, female, 70 yrs)

A couple of patients indicated that their symptoms also worsened following acute emotionally stressful events. In these cases, they felt not capable to control symptoms.

For me as the reader, the chosen quotation neither supports the above nor the below statement. If it does so, the authors don't succeed in explaining how, sufficiently, for me to understand.

The patient describes learning to say "no". Saying no however, can be a way to respect personal physical boundaries rather than emotional boundaries. Further the quote also suggests that saying no depends on other factors than possible symptom exacerbation. The patient describes how she evaluates potential activities based on possible personal gain and enjoyability. So it is not so much about avoiding "overstepping of boundaries", rather it is about gaining something, setting priorities.

Further, it is very interesting to learn how P12 has acquired the skills to say "no", -is she more

aware of what is good for her, is she less afraid of possible consequences...? However, the quote doesn't say anything about this – but maybe that could be an interesting theme on its own?

I don't have sufficient context to provide a meaningful analysis of a single quotation off course, so this should just be seen as inspiration.

The above general comments are relevant throughout the results section.

Below, I will present some more specific comments to individual subthemes.

Response: Thank you for your extensive commentary; this helped to strengthen our results section considerably. We revised this section thoroughly and checked all quotations. With regard to 'learning to say no': in an earlier version of the manuscript we included a theme focused on gaining control, in which we explored several strategies. This got lost over the various versions (and shortening) of the manuscript (there's a 4000 word limit). We agree it is however an interesting theme that was important in relation to symptom exacerbations, so we decided to discuss it in our revised version. Because we quite rigorously revised the results section, please see the manuscript for changes made, these are highlighted (pages 8-13).

P8: Theme 1 – the first section lists a number of quotations which support what you already knew from your sampling - that you have included patients who do in fact experience fluctuation. So this is hardly an interesting or surprising finding.

Response: We don't agree with this, what was surprising was that in the interviews there was no distinction in experienced fluctuations between the 'fluctuating' and 'seemingly stable' sample of patients. So, in that sense, it is quite surprising that both 'groups' of interviewed patients experienced fluctuations (over the day, week, longer periods). To emphasize the relevance of this finding, we added a sentence to underline this in the results section. Besides, we gave more emphasis to the 'two groups' ('fluctuators' and 'seemingly stable') in our sample in the methods section and in the discussion.

Changes (results, p. 8):

The following was added:

This meant that both the selected patients with fluctuations, as well as the seemingly stable patients in our sample experienced these fluctuations.

The quotations, however, also pertain to "patterns of fluctuation" – this might be a slightly different focus to explore in this theme?

Response: Good point. In a prior version of the manuscript, we actually spoke more about patterns (short-term fluctuations; long-term fluctuations) and we now decided to bring this back in the paper.

See amendments made on page 8 of the manuscript (results section).

P8 line 12: in your presentation of participant selection you emphasize the importance of selecting participants with fluctuating and stable symptoms respectively. Please elaborate how that relates to the fact that all participants in fact do experience fluctuation and how this may affect your findings.

Response: We indeed purposively sampled both patients who fluctuated as well as patients who seemed stable based on our quantitative findings. As mentioned above, our interviews actually showed no particular difference between the experiences of these two groups; fluctuations in symptoms were a central theme in the accounts of all interviewed patients. Therefore, although we anticipated differences, this was in fact not the case. In particular short-term fluctuations led patients to a continuous search to what caused exacerbations of symptoms to anticipate on exacerbations and prevent symptoms from worsening.

P8 line 21: paid or unpaid jobs, what do you mean? Does it matter?

Response: With unpaid jobs we meant volunteer work. What we meant is that, it didn't matter whether they had paid work or volunteer work. We decided to slightly change this formulation, because indeed it does not really matter.

Changes (results, p. 8):

"Worsening over the work week was described by all patients with (un)paid jobs."

Was changed into:

"Worsening over the work week was described by all patients who worked."

P 8 line 45: In these final quotes, patients talk about how the anticipation of symptom exacerbation, rather than experienced fluctuations, affect their daily life and activities. This may be a very relevant theme to explore. However, it does not fit in the title "experiences with fluctuation".

Response: We agree with your remark. When revising the results section, we replaced this section under theme 3 ('patients' strategies in gaining control over symptom exacerbations'), where we made a subheading named 'Adjusting daily planning'.

Changes:

Because we quite rigorously revised the results section, please see the manuscript for changes made, these are highlighted.

P9: Theme 2. This theme addresses the patient experience of why symptoms may fluctuate, which is very relevant. Maybe the title could reflect this. The introduction provides a very interesting and relevant conclusion and may be placed at the end of the subtheme description. The results have to support the claims though.

The section on "balancing physical limits" contains relevant quotes which support the statements.

The section on "crossing emotional boundaries" needs more work. As highlighted above, I cannot follow how quote 1 and 2 pertain to emotional boundaries.

Response: Thank you for your helpful remarks. We agree on adapting the title of theme 2 to better reflect its focus and changed it into 'perceived causes of symptom exacerbations'. We have critically revised the claims we made and removed concluding remarks and/ or claims suggesting causality from the results section. The section on 'crossing emotional boundaries' has been altered and is now

more focused on emotions and symptoms ('negative emotions'), which was an important (sub)theme in relation to the experience of why symptom may worsen.

Changes:

Because we quite rigorously revised the results section, please see the manuscript for changes made, these are highlighted in the results section.

P10 line 12: Quote 3 in this section is concerned with the association of negative emotions and symptom experience. I don't know if it is about emotional boundaries? However, the influence of negative emotions on symptom experience is extremely interesting and could be elaborated further.

Response: We agree. As mentioned in the prior response, we revised this section as mentioned above and discussed this in a subtheme about how patients linked symptom exacerbations to 'negative emotions'.

Changes:

Because we quite rigorously revised the results section, please see the manuscript for changes made, these are highlighted in the results section (p. 10+11).

P9: Theme 3: Dealing with fluctuations: experienced difficulties in balancing limits and boundaries

The theme is relevant and interesting; however, the introducing statement or conclusion again does not have sufficient support in the quotations and interpretations. Overall, I think that the quotes rather are concerned with the patients' ways of dealing with symptoms, priorities and life as such, rather than with dealing with fluctuations.

Response: We looked at previous versions of the manuscript and came to the conclusion that we should reintroduce strategies in gaining control over symptoms as a third main theme. So, we revised this entire theme and renamed it 'patients' strategies in gaining control over symptom exacerbations'. Because we quite rigorously revised the results section, please see the manuscript for changes made, these are highlighted in the results section.

P10: Lack of recognition and validation of symptoms

The experiences with lack of diagnosis are well described in the qualitative literature and an important issue to address.

Inspiration (maybe relevant in the discussion?):

- Salmon, P. (2007). Conflict, collusion or collaboration in consultations about medically unexplained symptoms: the need for a curriculum of medical explanation. *Patient Educ Couns*, 67(3), 246-254. doi:10.1016/j.pec.2007.03.008
- Dumit, J. (2006). Illnesses you have to fight to get: facts as forces in uncertain, emergent illnesses. *Soc Sci Med*, 62(3), 577-590. doi:10.1016/j.socscimed.2005.06.018
- Jutel, A. (2016). Truth and lies: Disclosure and the power of diagnosis. *Soc Sci Med*, 165, 92-98. doi:10.1016/j.socscimed.2016.07.037
- Jutel, A., & Nettleton, S. (2011). Towards a sociology of diagnosis: Reflections and opportunities. *Social Science and Medicine*, 73(6), 793-800. doi:10.1016/j.socscimed.2011.07.014
- Nettleton, S. (2006). 'I just want permission to be ill': Towards a sociology of medically

unexplained symptoms. *Social Science & Medicine*, 62(5), 1167-1178.
doi:<http://dx.doi.org/10.1016/j.socscimed.2005.07.030>

However, the claim made that lack of diagnosis/validation leads to overstepping of boundaries which again leads to symptom fluctuation is simply not one that can be supported by any qualitative data. Qualitative inquiry does not provide information about causation. Please revisit this section.

Response: Because we needed to shorten our manuscript due to the major revisions of our results section and because this is a well-known -though important- issue, we decided to highlight this only shortly in theme 3 'patients' strategies in gaining control over symptom exacerbations' under the subheading 'resigning to physical limits' (without making any claims about causation).

Changes:

Please see the manuscript for changes made on this part, these are highlighted in the results section (p. 12).

In the final section on page 11: Importance of resigning to limits and boundaries, the authors make statements about how patients have gained control over symptoms. Again, the quotes do not support the claims made. The final quote rather suggests that "not resigning into symptoms" is experienced as gaining control. The text does suggest so, however the overall conclusion you present later is that: overall – resigning is a good thing.

An interesting analysis might be the different perspectives on "gaining control" that you can find in your data. 1) "resigning", 2) "consciously overstepping", 3) "evaluating an activity and making choices based on personal preferences" (as was suggested in a previous quote). 4) I am sure you have a lot of data to provide perspectives on this matter. For the clinician, this could be very important information – gaining control is achieved differently by various patients, possible methods of gaining control were....

Response: Thank you for your helpful remarks. We looked at previous versions of the paper and went back to our data and came to the conclusion that we should indeed address strategies in gaining control over symptoms as a third main theme (as we also mentioned in prior responses) and extensively revised this part of the result section.

Changes:

Because we quite rigorously revised the results section, please see the manuscript for changes made, these are highlighted in the results section (p. 11-13).

Discussion

I will not comment on the discussion in too much detail, as it probably will change when the results section has been revised.

P11 line 16: The concluding remarks of the results are very interesting, however not sufficiently supported as described above.

Response: We revised our results, and part of these remarks became part of theme 3 (subtheme: resigning to physical limits), to support the statement we added a clarifying sentence and a quote.

Changes (results, p. 13), we added the following:

They for example encountered new situations as a result of changing environments and life changes over time.

"I still haven't fully embraced it and am not Zen about it. Because, you know, when I see other mothers. (...) Or when Mum plays tag or something. Then I run ten paces. Can't run too long, or I get myself in trouble. That still frustrates me." (P3, female, 32 yrs)

P11 line 47: I absolutely agree with the strength of and relevance of this study

Response: We are glad to hear you agree with the relevance of this study.

P12 line 13: This section starts with the claim that letting go of the search for a diagnosis results in less fluctuation which again results in improved wellbeing. Again, careful with claiming causation. Maybe you can find inspiration for the discussion of this point in the literature suggestions above.

Response: After our revisions of the results, we also revised the discussion section of our manuscript and made sure that any claims on a causal nature were taken out and the focus on patient experiences was highlighted.

Changes (discussion, p. 14+15):

Please see for changes made, the highlighted parts.

Checklists comments:

2. The abstract will need changes after the revision

Response: We revised the abstract and article summary after revising the manuscript.

Changes (abstract p. 2):

Because we quite rigorously revised the results section and discussion, the abstract has been altered quite extensive as well. Please see the manuscript for changes made, these are highlighted.

8. The references used are relevant and up to date. However, the list reflects focus on quantitative literature. More relevant qualitative literature might prove helpful.

A little more inspiration

• Moulin, V., Akre, C., Rodondi, P. Y., Ambresin, A. E., & Suris, J. C. (2015). A qualitative study of adolescents with medically unexplained symptoms and their parents. Part 1: Experiences and impact on daily life. *J Adolesc*, 45, 307-316.

doi:10.1016/j.adolescence.2015.10.010

• Nunes, J., Ventura, T., Encarnacao, R., Pinto, P. R., & Santos, I. (2013). What do patients with medically unexplained physical symptoms (MUPS) think? A qualitative study. *Ment Health Fam Med*, 10(2), 67-79.

• Karterud, H. N., Risor, M. B., & Haavet, O. R. (2015). The impact of conveying the diagnosis when using a biopsychosocial approach: A qualitative study among adolescents and young adults with NES (non-epileptic seizures). *Seizure*, 24, 107-113.

doi:10.1016/j.seizure.2014.09.006

Response: Thank you for these suggestions. In this new version we included different qualitative studies in our discussion section to broaden the perspective on our findings (discussion, p. 14+15 highlighted version).

9.-11. Please look to the comments on the results section above
I hope that you find the comments helpful and that they will allow you to revisit your very interesting study and data with new inspiration.
I want to thank you for allowing me to comment on your work.
Best wishes

Response: It has been really helpful to receive your extensive and very helpful commentary, which definitely helped to strengthen our paper.

Reviewer #3

Reviewer Name
MWF van den Hoogen
Institution and Country
Erasmus MC Rotterdam

Please state any competing interests or state 'None declared': None declared

Please leave your comments for the authors below:

The authors have performed an original and important study on a very common medical problem. Before their study it was indeed unknown how patients perceive the fluctuations in their symptoms and I find that my patients often try to find (sometimes in vain) any correlation between the fluctuations and any behavior or other attribute. Although the authors acknowledge that their data stems from a selected group of patients with PPS and might therefore not apply to all patients with PPS, I have learned a few points and so should other healthcare providers, to improve the care for these often 'difficult to treat' patients. That said, I have a few comments. First, in the methods selection both patients with fluctuations and those with (seemingly) stable course were included, however this distinction does not trickle down in the rest of the article. Can the authors explain why, was there no two different group of patients (if so, explain table 1 and this seems to contradict the first paragraph of the results, line 14-15 page 8) or did they not differ in their discussed themes? This could help personalizing the care for PPS patients.

Response: Thank you for your positive remarks. Regarding your question, we can understand that our results evoke some questions about the distinction between the patients we selected 'with fluctuations' and 'with a seemingly stable course'. The themes we identified were mentioned in both groups of patients– so indeed, there were no differences in the discussed themes and they both indicated fluctuations. The other themes were also identified in both groups. Because we probably did not clarify this well enough (Reviewer#2 also commented on this), we added some additional sentences on this in our results and discussion section.

Changes: The following was added (see highlighted version):

This meant that both the selected patients with fluctuations, as well as the seemingly stable patients in our sample experienced these fluctuations (results, p. 8).

Another possible limitation is that we had certain preconceptions based on our prior quantitative study and that these might have impacted our findings. We tried to minimize the impact of our preconceptions by our theoretical sampling method in which we included both -'fluctuating' and 'seemingly stable' patients. Although we anticipated differences in experience between these patients, this was in fact not the case. (discussion, p. 14).

More-over I would like to see more details about the patients. What kind of PPS do they have? The authors state it is over more domains, but to give a clear picture to readers, it would help to see that xx% have pain, xx% have fatigue, etc etc.

Response: Thank you for your suggestion, we added more details on the experienced symptoms to table 1.

Changes to table 1, page 6:

We added the following:

Symptoms

-Fatigue	(12/15)
-Musculoskeletal pain	(12/15)
-Headache	(6/15)
-Gastro-intestinal symptoms	(5/15)
-Cardiopulmonary symptoms	(3/15)
-Dizziness	(3/15)

Furthermore, can the authors give more quantification on the level of fluctuation over both long and short term. Are they really fluctuations or only perceived as such, if real, how much and is that amplitude of fluctuation related to outcome e.g. are high fluctuators more impacted on daily life than low fluctuators.

Response: This is a difficult point to address in a qualitative study like this. Because we aimed at experiences -which is of course reality in the experiences of people- and although we used theoretical sampling to interview patients with fluctuations and as well as seemingly stable patients (based on theory)- we did not compare their accounts with their quantitative findings over time. As we also indicated in our prior response, fluctuations were mentioned to the same extent by both patients with a 'seemingly stable course' as well by 'high fluctuators'. It should be taken into account that 'seemingly stable' patients were rare in our cohort (<15%, it may in fact be coincidence that they seemed stable...), we added an additional sentence about this to the methods section.

Changes (box 1, page 5):

Over a three years' period, only a minority of the participants (<15%) showed clinical stability in symptom severity and physical functioning.

Another point might be about the external validity. Can the authors elaborate more how applicable these results might be in a culturally different population than Dutch people, since I think culture and perception of PPS are two very important issues. Can we use these data on patients from other cultures living in the Netherlands / Western Europe?

Response: Each qualitative study is unique in its setting and cases. Therefore, the term external validity is not applicable here as it is in quantitative studies. So, although there may be some transferability to other settings, this remains subjective and transferring these findings to other cultures living in the Netherlands/ Western Europe would be presumptuous. It would be useful to repeat this study in patients groups with other cultural norms.

Can the authors also give 1-2 extra examples on the topic of “importance of resigning to limits and boundaries”?

Response: Because of major revisions in our results section on this, which were based on comments of reviewer #2, we amended this part of the manuscript. In theme 3 we now more extensively discuss this in the subtheme ‘resigning to physical limits’ with some additional quotes.

Changes:

Please see the manuscript for changes made, these are highlighted in the results section (p. 12).

Finally, I am not sure whether all readers have the same understanding of the term ‘resignation’ which is used frequently. I suppose the authors use a different wording or explain the term.

Response: Thank you for your comment, we understand that this term might be interpreted differently by readers – in fact we used the Dutch word ‘berusten’ (we understand you are Dutch as well) and asked a professional translator how to translate this, because we did not know what best describes this, as it is different to ‘accepting’. The translator (also based on the context) advised us to use ‘resignation’. Nevertheless, we understand your concern and added an explanation to the term used and added an extra quote to illustrate this.

Changes (results, p. 12-13):

The following was added:

By resigning, we mean that patients expressed the need to take their physical limits seriously and anticipate by limiting their activities in order to prevent exacerbations of symptoms. Resigning to limits was experienced as different from accepting their limits, as many kept struggling with the acceptance of their physical limits. They for example encountered new situations as a result of changing environments and life changes over time, again confronting them with their physical limits.

“I still haven’t fully embraced it and am not Zen about it. Because, you know, when I see other mothers. (...) Or when Mum plays tag or something. Then I run ten paces. Can’t run too long, or I get myself in trouble. That still frustrates me.” (P3, female, 32)

In all I think the authors will contribute substantially to the medical field of PPS / MUPS if these data are published.

Response: Thank you for your feedback; it definitely helped us in strengthening our manuscript.

VERSION 2 – REVIEW

REVIEWER	Ditte Roth Hulgaard University of Southern Denmark Denmark
REVIEW RETURNED	21-Apr-2020

GENERAL COMMENTS	I want to congratulate the authors with this much-improved manuscript! I recommend this manuscript to be accepted and have only 2 very minor suggestions for the authors: P 7 line 6 – 7. Please specify what the abbreviations for the questionnaires stand for. P13 line 57: After carefully describing preconceptions and how you dealt with them, I would not consider them to be a limitation, so you might consider deleting this limitation :)
--

REVIEWER	MWF van den Hoogen Erasmus MC, Rotterdam
REVIEW RETURNED	18-Apr-2020

GENERAL COMMENTS	The authors have addressed my concerns in a correct manner and i have no additional questions
---

VERSION 2 – AUTHOR RESPONSE

Reviewer' Comments to Author:

Reviewer: 2

Reviewer Name

Ditte Roth Hulgaard

Institution and Country

University of Southern Denmark
Denmark

Please state any competing interests or state 'None declared':
None declared

Please leave your comments for the authors below: I want to congratulate the authors with this much-improved manuscript!

-I recommend this manuscript to be accepted and have only 2 very minor suggestions for the authors:
P 7 line 6 – 7. Please specify what the abbreviations for the questionnaires stand for.

Response: Thank you for your positive remark and helpful suggestions. We specified the abbreviations for the questionnaires.

Changes:

-Methods, page 8: we added the highlighted text in the following sentence:

“For the PROSPECTS study, patients filled in questionnaires about the nature and severity of their symptoms (Patient Health Questionnaire-15 (PHQ-15), 0-30 scale (20)) and physical functioning (RAND-36 Physical Component Summary (PCS), 0-100 scale (21)) among other questionnaires.”

-P13 line 57: After carefully describing preconceptions and how you dealt with them, I would not consider them to be a limitation, so you might consider deleting this limitation :)

Response: We agree with the reviewer that indeed our preconceptions - after carefully describing our preconceptions and how we dealt with them- were no longer a limitation and decided to delete our preconceptions as a limitation of our study from the discussion section.

-Discussion, page 11-12: we deleted the following sentence from the discussion section of our manuscript:

“Another possible limitation is that we had certain preconceptions based on our prior quantitative study and that these might have influenced our findings.”

and slightly changed the order of the sentences of the paragraph.